# Fertility Preservation and Reproductive Potential in Transgender and Gender Fluid Population

**DOI:** 10.3390/biomedicines10092279

**Published:** 2022-09-14

**Authors:** Ji Young Choi, Tae Jin Kim

**Affiliations:** 1Department of Gynecology and Infertility Medicine, CHA University Ilsan Medical Center, Goyang 1205, Korea; 2Department of Urology, CHA University Ilsan Medical Center, CHA University School of Medicine, Goyang 1205, Korea

**Keywords:** fertility, fertility preservation, transgender

## Abstract

The gender diverse and transgender community is a minor patient group that is encountered with increasing frequency in the clinical setting, attributed to the improved awareness and access to medical facilities. Partial impairment to permanent elimination of fertility potential and outcomes depending on the treatment modality usually is a result of gender-affirming therapy, which includes both hormone therapy and surgical intervention. Although seldom conducted in the clinical field, transgender patients should be counseled on their fertility preservation options prior to medical and surgical gender transition. There is relatively limited data and clinical information regarding fertility preservation for transgender individuals. Current treatment regimens are based on protocols from fertility preservation after oncological treatments. Major barriers for the transgender population exist due to the lack of information provided and clinical narrative that is not familiar to the physician or health care provider, although there are various options for fertility preservation. A deeper understanding of this clinical agenda and the mandatory processes will ultimately result in a much more comprehensive and specific care for transgender individuals who are in great need for fertility counseling or treatment options that concern fertility preservation. In this review, current clinical approaches will be summarized and fertility preservation options along with ongoing and future clinical trials in fertility preservation for transgender individuals will be thoroughly reviewed.

## 1. Introduction

The transgender community represents a minor but growing patient population in the field of fertility and reproductive medicine. Transgender and nonbinary individuals comprise approximately 2% of the total population, with an estimated total number of 1.4 million transgender and nonbinary people residing in the United States, alone according to a 2016 survey [1]. The need for fertility preservation counseling for the transgender and nonbinary patient population has been advocated by various medical associations, such as the Endocrine Society and the World Professional Association for Transgender Health (WPATH) [2]. The majority of transgender individuals experience a difference in gender identity from their biologically assigned sex during puberty with a mean age of presentation at 27–32 years [3]. Definitely, there is a growing need for gender-affirming treatment and a trend toward a decrease in initial age at the time of presentation [4].

Access to medical treatment and hormonal interventions has become less challenging to this minority population with an increased cognizance of gender variance and social acceptance to transgender and nonbinary people. Gender-affirming hormonal therapies include puberty-suppressing gonadotropin-releasing hormone agonists (GnRH) and sex hormone treatments, such as estrogen administration for those desiring feminine secondary sex characteristics and testosterone hormone therapy for male secondary sex characteristics. Surgical interventions of gender-affirming therapy consist of breast augmentation or mastectomy and gonadectomy. Although treatment modalities are potentially available to all gender fluid and transgender individuals, the patient may undergo some or all of these aforementioned treatment modalities, depending on their specific needs [5].

With the increase in gender-affirming therapy, counseling transgender patients about fertility and fertility preservation has become an area of increasing concern. One of the main adverse effects in the long-term use of gender-affirming hormone therapy is infertility, although further clinical studies are required to elucidate the exact duration of hormonal treatment that causes irreversible effects [6,7]. Recent studies show that approximately 50% of transwomen adults and 37.5% of transmen individuals would have considered gamete freezing prior to gender-affirming medical therapy if they were informed [8,9]. The importance of fertility counseling and informed consent for gamete preservation prior to gender-affirming hormonal treatments has been stipulated in the American Society for Reproductive Medicine (ASRM), Endocrine Society, and WPATH clinical guidelines [2]. Despite the increasing numbers of fertility preservation counseling, recent studies based on the pediatric and adolescent transgender population report a <5% rate of gamete preservation [10,11]. Moreover, in a study conducted by Chiniara and colleagues, none of the 79 transgender patients underwent fertility preservation prior to gender-affirming treatments [12]. However, higher rates of fertility preservation were seen in transfeminine adolescents in the Netherlands where approximately 35% of the study cohort initiated fertility preservation [13].

For this specific patient population, physicians and medical providers may be less accustomed to the treatment modalities that are available. Results from a 2019 survey led by Tishelman et al. showed that medical providers from different nationalities acknowledged the need for further data and research based on the available options for transgender fertility preservation [14]. There is a restricted amount of clinical data specific to fertility preservation for transgender individuals, and the majority of contemporary treatments are based on options for fertility preservation reserved for patients with oncologic diagnoses.

In this review, current clinical approaches and options on fertility preservation for the transgender population will be reviewed and provide insight along with an in-depth discussion on ongoing trials and future studies on fertility preservation for transgender individuals.

## 2. Understanding Gender Identity and the Parenting Desire of the Transgender Community

### 2.1. Definitions of Gender Identity

The spectrum of an individual’s specific gender identity varies considerably in the transgender population. Often described as nonbinary, the gender of the transgender population falls outside the traditional binary system of man/woman. Commonly used terms to define an individual who does not fit the gender binary classification include gender fluid, gender neutral, and gender diverse. Gender identity is a person’s initial experience of gender or the internal being of self from a gender perspective [15,16]. Gender dysphoria is the condition of psychological distress associated with gender identity and biologically assigned sex [16]. Cisgender refers to an individual whose gender identity is consistent with the sex assigned at birth, whereas transgender people have discrepancies in their gender identity and physically assigned sex [17]. Transwoman individuals are those who were assigned male at birth but identify as female, whereas transman individuals are those who were assigned female at birth but identify as male. Additionally, there is increasing recognition of gender fluid people who do not have a fixed gender identity [17].

### 2.2. Parenting Desire and Family Building

Many patients have shown their interest in having biological children, contrary to the prevalent assumption that transgender people are not interested in parenthood. In some studies, more than half of the patients showed a strong desire to become parents; one study showed that 54% of transmen desired to have children of their own [9], whereas another showed that 51% of transwomen would have undergone fertility preservation via sperm cryopreservation had they been informed prior to gender-affirming treatment [8]. Transgender people who have had gender-affirming treatment at a relatively advanced age have a tendency to become parents compared to those in younger age groups [8]. Moreover, a large number of the transgender community report an interest in building a family via adoption [10]. In a recent study based on adult transgender people who had undergone gender-affirming therapy and identified as having a desire to become future parents, approximately one-third of the study population considered adoption for family building [9]. In addition, adolescent transgender people believe adoption as a viable option for family building [11]. Nahata et al. based a study on transgender teenagers who were counseled prior to gender-affirming therapy and 45% of the population refused fertility preservation due to plans to adopt children in the distant future [11]. As awareness regarding fertility preservation seems to grow with easier access to fertility treatments and fertility counseling of preserving gametes before possible irreversible gender-affirming treatment, further studies are needed to optimize treatment protocol and guidelines for transgender patients who have a parenting desire.

## 3. Gender-Affirming Therapy and Fertility Preservation Methods for Transmen

### 3.1. Gender-Affirming Hormone Therapy for Transmen

Testosterone is the treatment of choice for transmen and nonbinary persons undergoing masculinizing therapy. Testosterone is administered to transmen who are receiving masculinizing gender-affirming medical treatment via intramuscular/subcutaneous injection or through a transdermal route. Testosterone administration causes amenorrhea in the majority of the patients, as shown in one study where 82% of the patients became amenorrheic within 3 months [18]. However, hormone therapy does not necessarily stop ovulation, and there have been case reports of transmen conceiving during testosterone treatment [19]. Therefore, physicians should warn patients undergoing testosterone therapy of possible pregnancy and advocate contraception. 

Histopathologically, testosterone administration affects folliculogenesis and the ovaries. Grimstad and colleagues showed that testosterone therapy for a mean exposure time of 36 months resulted in persistent ovarian function with follicular cysts in approximately half of the study population (49%), polycystic ovaries in 6%, and ovarian atrophy in 4% of the cohort, whereas 39% of the patients had normal ovarian histology. Patients with prolonged testosterone treatment showed an overall decrease in ovarian volume, although the mean volume of the ovaries in the study population remained within the normal range [20]. In a recent prospective study, histopathological analysis using fluorescence-activated cell sorting was performed with the ovaries of transmen receiving gender-assigning hormone therapy. Results from the study showed that 88% of the ovarian cells showed normal cortical follicle distribution, which shows a disparity to previous pathological studies where most of the ovarian specimens had polycystic or atrophic attributes [21]. Adverse events and the effects of long-term testosterone exposure to ovarian tissue are still under study and research. To further elucidate the physiological impacts of long-term testosterone administration and exposure, future prospective clinical trials and studies are required.

### 3.2. Fertility Preservation for Transmen

Fertility preservation options for transgender men are oocyte cryopreservation, embryo cryopreservation, and ovarian tissue cryopreservation with in vitro maturation (IVM) (Table 1). Oocyte or embryo cryopreservation involves invasive procedures, such as controlled ovarian stimulation (COS) cycles which require about 2 weeks of daily gonadotropin injections to induce hyperstimulation of ovaries followed by oocyte retrieval under anesthesia. Some distressing aspects of ovarian stimulation include physical examinations including transvaginal ultrasonography, discontinuation of testosterone administration, and resumption of menstruation [22]. Patients who decide to begin COS for gamete preparation should be advised about the increase of serum estradiol levels due to hyperstimulation and the distressing nature of serial transvaginal ultrasonography for assessment. To encourage and enable more transmen to have COS, the addition of aromatase inhibitors to minimize the increase in serum estradiol levels and replacing transvaginal ultrasonography exam with transabdominal ultrasonography are some alternative methods that could be integrated into the clinical setting [23]. Clinical studies based on oocyte and embryo cryopreservation in transmen patients have reported promising results. A retrospective cohort study of transgender adolescents showed more comparable outcomes in both quality and quantity of oocytes than the cisgender group [24]. Moreover, a retrospective study involving 26 transgender men showed promising fertility preservation outcomes with a mean of 20 oocytes retrieved. Among the enrolled patients, seven patients had their embryos transferred, followed by a successful pregnancy [25]. During fertility preservation counseling, a major issue is planning for the gestational carrier and future embryo. There are reports of transmen who elect pregnancy either because of partner infertility or due to personal desire to conceive [15]. However, others refuse reimplantation into their own uterus due to gender dysphoria and incongruity to their gender identity [16]. Transmen who have completed both gender-affirming hormone and surgical treatment which include hysterectomy and oophorectomy are completely sterile. Further and detailed fertility counseling regarding gender dysphoria during pregnancy, obstetric monitoring, and postpartum events, which include resuming testosterone administration and infant care, should be completed [19,25].

### 3.3. Oocyte Cryopreservation

There are few studies and limited data concerning COS while on gender-affirming hormonal treatment. Theoretically, a prolonged testosterone discontinuation may result in the reoccurrence of intrinsic ovarian stimulation and a healthier resting follicular pool compared to those on hormone therapy. However, the majority of the transmen community is reluctant to the discontinuation of hormonal treatment, resulting in a lower participation rate in fertility preservation. One study confirmed that a patient with a 1-week cessation of testosterone treatment was able to undergo ovarian stimulation and ended up with successful cryopreservation of 11 mature oocytes [26], whereas a recent case report showed that a transman who had been on long-term testosterone treatment without discontinuation underwent COS and has successfully cryopreserved 22 oocytes [27]. Additionally, Stark et al. reported two cases of fertility preservation where patients had been on testosterone therapy for 6 and 20 months and underwent COS without discontinuation of hormonal therapy, leading to 30 and nine metaphase II (MII) oocytes successfully cryopreserved, respectively [28]. These studies show that the collection of oocytes is indeed possible despite the presumption that testosterone might have a negative impact during such course of treatment, although the quality of oocytes, pregnancy outcomes, and long-term follow-up outcome of children born through oocytes retrieved from ovarian stimulation after short-term discontinuation and even continuation of testosterone treatment are yet to be discovered [29]. The optimal age group and timing for oocyte cryopreservation is another aspect that should be thoroughly covered. When considering the different oocyte aneuploidy rates in relation to aging, the timing of oocyte cryopreservation is of utmost importance. Cobo et al. reported clinical pregnancy outcomes of cisgender women who have had fertility preservation via oocyte cryopreservation. Among ciswomen who belonged to the younger age group of less than 35 years of age, the cumulative live birth rate was as low as 15.4% when only five oocytes were used. However, when eight or ten oocytes were utilized, 40.8% and 60.5% of live birth rates had been reported, respectively. In contrast, among ciswomen who belonged to older than 35 years of age group, the cumulative live birth rates were 5.1%, 19.9%, and 29.7% with five, eight, and ten oocytes, respectively [30]. Various studies have shown that not only quantity but also quality of oocytes decline as a result of the aging process, and results show that the ideal time for oocyte cryopreservation is before age 35 [31,32]. Franasiak and colleagues revealed that the lowest aneuploidy rate was in those aged 26 years to 37 years, whereas the prevalence of aneuploidy went up to as much as greater than 40% in those younger than 23 years of age [33]. The aneuploidy rate in oocytes showed a U-shaped curve throughout the lifetime, and those younger than age 13 years and older than mid-30 years have a reduced rate of conception [34]. Patients who fall into the age group of younger than 23 years and older than 35 years of age should have in-depth counseling about the likelihood of the actual usage of the gametes and possible future outcomes due to concerns of aneuploidy [29].

### 3.4. Embryo Cryopreservation

As an alternative fertility preservation method, transmen patients have the option of embryo cryopreservation. Patients may use cryopreserved oocytes with sperm from either a partner or a donor for embryo fertilization. One of the advantages of cryopreserved embryos when compared to oocyte cryopreservation is the confirmation of higher quality blastocysts developed from the retrieved oocytes. Furthermore, several studies have reported a higher survival rate of the embryos (95–98%) in comparison to the 80% survival rate for oocytes [30,35]. Embryos are eligible for preimplantation genetic testing for aneuploidy unlike ovarian tissue or oocytes. Due to the requirement of donor sperm or male partner and the need for dual consent before the implantation of the embryos, the main disadvantage of embryo cryopreservation is the lack of autonomy and flexibility in making future decisions [36].

### 3.5. Ovarian Tissue Preservation with IVM

In 2019, the ASRM practice committee announced that ovarian tissue cryopreservation with IVM is no longer considered experimental [37]. This fertility procedure enables oocyte development from the germinal vesicle stage to the MII stage, which is suitable for fertilization. Sample preparation is performed via surgery during which all or part of the ovary is removed and strips of ovarian cortex are cryopreserved [38,39]. Ovarian tissue cryopreservation may be performed during gender-affirming surgery without undergoing hormonal change which may help avoid gender dysphoria. In transmen with a male partner, ovarian tissue can be a potential source for IVM with mature oocytes fertilized by partner sperm along with a gestational carrier. For transmen with a female partner, IVM can be done with mature oocytes fertilized with donor sperm and the embryo is transferred to the partner’s uterus. A clinical study analyzed 680 cumulus-oocyte-complexes obtained from 16 transmen after prolonged administration of testosterone prior to gender-affirming surgery, and 38% of the cumulus-oocyte-complexes were found to be mature and showed normal spindle structure. This finding suggests that testosterone treatment does not yield morphologic effect on oocytes [40]. Another advantageous aspect of ovarian tissue cryopreservation is that it is the only available means to preserve their own gametes for prepubertal transgender patients who seek fertility preservation. Clinical outcome using ovarian tissue cryopreservation has shown potential, where cisgender women have had more than 130 live births using cryopreserved ovarian tissues [41,42]. However, since clinical studies have reported various outcomes, the fertility potential of IVM is still under evaluation. Lierman et al. evaluated the quantity and quality of the blastocysts formed from cryopreserved ovarian tissue, where out of 208 mature oocytes retrieved from cryopreserved ovarian tissues, only two blastocysts of good quality were confirmed. The authors concluded that the reasons for unfavorable outcome are poor oocyte maturity, difficulty in fertilization, abnormal cleavage formation, and arrest of embryo development [40]. Moreover, various studies have shown comparable pregnancy outcomes of cryopreserved ovarian tissue that belong to early pubertal adolescents [43,44,45]. Further research and clinical studies are needed to overcome the setting where autologous transplantation of ovarian tissue into the pelvic cavity is required, which is undesirable for the majority of transmen [46].

## 4. Methods for Transwomen

### 4.1. Hormone Therapy for Transwomen

Having female secondary sex characteristics and serum hormonal levels of estrogen and testosterone similar to cisgender women is the main objective of gender-affirming hormonal treatment in transwomen. Feminizing hormone therapy for transwomen usually involves a combination of androgen-lowering medications, which include GnRH agonists, spironolactone, or cyproterone acetate along with estradiol [47]. Feminizing hormonal therapy resulted in the suppression of spermatogenesis which can be confirmed by pathologic analysis of testicular tissue [48,49]. High doses of estrogen treatment have been shown to decrease sperm count within 2 weeks, whereas prolonged application of hormonal therapy resulted in spermatogenesis suppression and regression of Leydig and Sertoli cells into immature precursors [50,51]. A recent histopathologic study of testicular tissue from a large prospective cohort of transwomen, where 97 transwomen underwent gender-affirming hormone treatment, showed no evidence of spermatogenesis in 77% and partial spermatogenesis in 23%, and complete spermatogenesis was absent in the total study population [52]. To further elucidate the pathological influence of feminizing hormone therapy on spermatogenesis, future research is mandatory, whereas further studies are needed to identify potential risk factors for hormone-naïve transwomen undergoing gender-affirming hormone treatment.

### 4.2. Fertility Preservation Options for Transwomen

Semen cryopreservation or testicular tissue cryopreservation is one of the fertility preservation options for transwomen (Table 2). Sperm cryopreservation is a feasible option for post-pubertal patients and is recommended prior to gender-affirming hormone treatment initiation. An analytic study based on sperm quality in transwomen who had sperm cryopreservation done showed that in comparison to transwomen continuing hormonal therapy, better semen parameters were observed in patients who discontinued gender-affirming hormones. Transwomen who underwent sperm cryopreservation prior to hormonal therapy showed a mean motile sperm count of 63.2 M, whereas 39.1 M of motile sperm was observed after 3 to 6 months of discontinuation of the hormone; the sperm count was diminished (0.2 M) in transwomen who continued hormonal therapy (*p* < 0.01). The authors of this study concluded that feminizing hormone effects on spermatogenesis may not necessarily be irreversible [53]. A recent prospective cohort study analyzed sperm cryopreservation samples in transwomen prior to gender reassignment treatment and the study revealed that the study population had worse sperm parameters when compared to the World Health Organization reference population [54]. Another analysis of cryopreserved sperm samples collected from transwomen before gender-affirming treatment showed that only 26% of thawed semen were adequate for intrauterine insemination [55]. The decreased serum parameters among treatment-naïve transmen imply that other risk factors for the impairment of spermatogenesis are present and further in-depth studies are needed to have a better understanding of underlying mechanisms of hormonal treatment on spermatogenesis in transwomen and other potential risk factors for relatively poor sperm parameters in transwomen prior to hormonal therapy.

Many transwomen patients find masturbation or ejaculation distressing, resulting in worsening of gender dysphoria, although it may seem relatively less invasive compared to transmen who have to undergo ovarian stimulation and oocyte retrieval. Some patients may have erectile and/or ejaculatory dysfunctions due to hypoandrogenism and electroejaculation, and penile vibratory stimulation may be helpful in such cases [15,56,57]. To retrieve sperm for patients who prefer to avoid ejaculation and masturbation or in patients who have been identified as oligozoospermic or even azoospermic, testicular sperm aspiration or microsurgical testicular sperm extraction are alternative methods [46]. Currently, testicular tissue cryopreservation is the only available fertilization preservation method for prepubertal transgender girls (Table 2). To elucidate outcomes and adverse effects, patients should be warned of the experimental nature of the procedure as further research is required. Future usage of cryopreserved testicular tissue requires auto-transplantation to the testis, scrotum, or ectopic subdermal anatomical sites [58,59,60]. Although the aforementioned procedure is considered experimental, various studies have reported promising results. Animal studies are underway to develop techniques to turn spermatogonial stem cells to mature sperm that are adequate for conception and fertilization [61]. A recent animal study involving the rhesus model has shown complete spermatogenesis after re-implantation, which resulted in both fertilization and live birth [62]. Limited data are available regarding the virility of human sperm from autologous transplantation tissues and there have been no study results reporting successful live birth with this method, despite such progress in animal studies [63].

Transwomen who have completed gender-affirming reconstruction are irreversibly sterile, and there are no fertility preservation methods available at this final stage of gender-affirming treatment. Alternative options such as surrogacy or uterine transplantation can be considered for this group of patients. Due to the diverse nature of elected partners in the transgender and gender fluid population, it is necessary to consider the fact that a gestational carrier may be required for a transgender couple to build a family [41]. Uterine transplantation is a potential option and deceased donors would be preferable, since vaginal and uterine transplantation could be achieved. The procedure is still in the experimental stage and has various medical risks and should be considered as a last resort and not an alternative means to a gestational carrier. However, with continuous research and advances, uterine transplantation has the potential to be a viable fertility preservation option in the future [64,65]. 

## 5. Ongoing Studies and Future Clinical Trials

The Micro RNA Profile in the Ovarian Follicle Fluid of Transgender Men Treated with Testosterone and the Association with Fertility Potential (NCT03725280) [66] is a prospective case-control observational study focused on the outcomes of testosterone exposure in fertility of transgender men. The effects of testosterone treatment in transgender men, particularly on the oocyte and embryo development, are yet to be elucidated. The objective of this study is to gain information about the effects of testosterone treatment in transmen on the oocytes and embryos during the in vitro fertilization (IVF) process, given that the extracellular ribonucleic acids (RNA) from the ovarian follicle cells represent the biological status of the oocyte and embryo. Within a 5-year timeframe, the primary outcome includes characterization of the micro RNA profile in the follicular fluid of transmen who have been treated with testosterone. The collection of the follicular fluid of transmen will be performed during the oocyte retrieval procedure, and micro RNA will be isolated from the follicle fluid and sent for RNA sequencing analysis via RNA library preparation kits, whereas total serum testosterone levels (ng/mL) will be analyzed in all patients prior to IVF. November 2023 is the estimated study completion date.

A Belgian observation retrospective study entitled “Fertility preservation in Transgender Persons: A Retrospective Look at the Decision and a Survey of the Desire for Children” (NCT05120245) [67] is currently ongoing. With an estimated enrollment population of 100 participants, a validated questionnaire will be used to survey the patients about how they feel about the decision for gamete cryopreservation. The primary outcome will be based on the decision along with the intention to use the frozen gametes. This study mainly focuses on practical aspects of sperm/oocyte cryopreservation and not on experimental techniques such as cryopreservation of ovarian and testicular tissue. 

”Fertility preservation and Reproductive Needs of Transgender People: Desires, Attitude, and Knowledge of Subjects with Gender Dysphoria” (PaFer) (NCT03836027) is a prospective cohort observational study that has completed recruitment [68]. The aim of the study was to describe desires, attitudes, and knowledge of transgender people regarding fertility preservation and reproductive needs based on the fact that there is limited knowledge based on fertility preservation in the transgender population. A cross-sectional anonymous survey was used for the 100 participants that were enrolled in the study. The survey includes general information on the characteristics of respondents, knowledge on fertility preservation, parental desires, and constraints and attitudes regarding fertility preservation issues. Secondary and further outcomes of the study will assess the knowledge of options of fertility preservation, general quality of life, state of anxiety, and depression.

The recently completed Fertility Decision Making in Youth and Young Adults (AFFRMED) (NCT05175170) [69] is a single-arm, pre-/post-feasibility, acceptability, and preliminary efficacy trial. Ten transgender young adults and 10 parents of transgender young participants were enrolled in a 90–120-min session of virtual research and videoconferencing. During this virtual visit, the participants are evaluated regarding fertility knowledge and decisional self-efficacy before and after the free 1 h navigation of the web-based AFFRMED decision aid prototype. The decision aid entails Internet domains based on human reproduction, fertility, and preservation. The learning objectives of each domain are to increase knowledge on the effects of fertility preservation and gender-reaffirming treatments. The primary study outcome is the change in fertility-related knowledge from pre-AFFRMED to post-AFFRMED exposure. Other endpoints of the study consist of feasibility, acceptability, and appropriateness of AFFRMED. Table 3 summarizes ongoing studies that focus on fertility issues in the transgender population.

## 6. Limitations and Difficulties in Fertility Preservation

Although there is a strong desire for parenting among transgender individuals, various studies have reported the low usage of available reproductive and fertility services. Riggs et al. conducted a survey with a study population of 409 transgender and nonbinary individuals and results showed that of the people who were not parenting, 33% of the cohort had a strong desire to be parents [70]. Moreover, the need for counseling and access to fertility preservation for transgender and nonbinary people was advocated by the majority of the study group [68]. Additionally, a public opinion survey showed that 76.2% of respondents agreed to the availability and access to medical treatment so that transgender individuals could have children of their own [71]. In a fairly recent German study, results showed that transmen had a significantly higher parenting desire when compared with transwomen prior to gender-affirming treatment. However, the study showed that an equal desire in parenting was present in approximately 25% of participants of both genders who underwent gender-affirming treatment [72].

There is still an inconsistent and a small percentage of patients utilizing fertility preservation counseling and procedures, despite recommendations for fertility preservation counseling prior to gender-affirming medical treatments. Chen and colleagues showed that with transgender adolescents initiating gender-affirming hormone therapy and were referred for fertility preservation counseling, approximately 12% had preservation counseling, and only 5% of the study group underwent gamete cryopreservation [10]. Only 45% of the participants had future parenting plans such as adoption whereas 21% of the study group had no desire for parenting as described in another study based on transgender adolescents [11]. According to a study done by Wierckx et al., in the majority (77%) of transmen who had gender-affirmation surgery, 77% had not been counseled or considered fertility preservation prior to gender-affirming treatment. Moreover, more than half of the study population had a strong parenting desire and transmen with children reported a significantly higher quality of life [9].

The transgender community experiences various limitations to achieving parenthood such as cost of treatment or lack of insurance coverage, gender dysphoria, and the invasive nature of gender-assigning procedures, although there are established recommendations for fertility counseling prior to gender-assigning therapies [10]. A Korean survey study concluded that financial and monetary factors were the main cause of hormonal therapy discontinuation in transgender patients [73]. Similar results were observed in the transgender community from the United States. Abern et al. showed that approximately half of the study population neglected fertility preservation due to the expensive nature of medical costs [74]. The aforementioned studies reflect the inconsistency of fertility preservation of transgender people and that the mandatory medical standards are not up to par. Clinical trials and studies have shown the potential adverse effects of gender-assigning treatments, and the need for adequate medical counseling and financial aid for the transgender community should be advocated.

The invasive nature of fertility preservation procedures is another barrier that causes reluctance for transgender patients to continue treatment, in addition to the high medical costs. Ovarian stimulation may take several weeks with multiple injections of gonadotropins, which results in the increase of serum estradiol levels, and the transgender patient is exposed to multiple transvaginal ultrasounds which may trigger gender dysphoria [75]. The amount of clinical data and information with which to advise transgender men who are receiving testosterone about ovarian stimulation for fertility preservation are limited. Moreover, current protocols for the duration of testosterone administration prior to ovarian stimulation are nonexistent, and no standard stimulation guidelines are available for patients undergoing testosterone therapy. Similar concerns exist for transwomen, as sperm collection methods may also exacerbate gender dysphoria and a reminder of their sex assigned at birth.

There are currently no guidelines regarding fertility preservation for transgender individuals. As the parenting desire and demand of fertility counseling of transgender people is increasing, counseling and medical services should be readily available [70]. Since the effects of gender-assigning treatment on reproduction are still under study, transgender patients were recommended to undergo fertility preservation prior to the initiation of gender-assigning therapy. However, the number of younger and adolescent transgender patients are increasing and the parenting desire of this population are yet to be defined and have the potential to change over time. Fertility counseling and gender-assigning treatment have to work simultaneously and in concert; however, the limited amount of information regarding the hazards to reproductive function and fertility-related outcomes of gender-assigning medical treatments confines the effectiveness of fertility counseling. Future clinical trials and prospective studies based on the risks of gender-assigning treatment along with fertility protocols and outcomes are needed to further address these limitations.

## 7. Conclusions

Results from various studies and clinical trials based on gender-affirming therapy have shown a detrimental impact on the potential for future fertility. Transgender patients undergoing gender-affirming treatment are at a risk for decreased reproductive capacity, and there is a growing awareness in the medical community focusing on the need for fertility preservation counseling prior to initiation of gender-affirming care. However, numerous studies have reported a limited standard of care with many transgender patients who are experiencing inadequate fertility counseling. Further prospective multicenter studies are needed to address the aforementioned shortcomings and advocate for better quality fertility services and parenting options for transgender and nonbinary individuals.

## Figures and Tables

**Table 1 biomedicines-10-02279-t001:** Fertility preservation options for transmen.

Fertility Preservation Method	Protocol	Advantages	Limitations
Oocyte cryopreservation	COS cycles requiring about 2 weeks of daily gonadotropin injections to induce hyperstimulation of ovaries followed by oocyte retrieval under anesthesia	Well-established method No need for partner Autonomy over gametes	Post-pubertal patients only Pelvic exams, transvaginal ultrasonography required Cessation of gender-affirming hormonal therapy recommended Menstruation may resume Hormonal treatment may lead to gender dysphoria Invasive method involving anesthesia Lower survival rate compared to that of embryos
Embryo cryopreservation	COS cycles requiring about 2 weeks of daily gonadotropin injections to induce hyperstimulation of ovaries followed by oocyte retrieval under anesthesia	Well-established method Confirmation of the quality of embryos developed from retrieved oocytes (good quality embryos have higher survival rate) Preimplantation genetic testing to check for aneuploidy	Post-pubertal patients only Pelvic examination, transvaginal ultrasonography required Cessation of gender-affirming hormonal therapy recommended Menstruation may resume Hormonal treatment may lead to gender dysphoria Invasive method involving anesthesia Lack of autonomy due to need for sperm (from partner or sperm donor) and dual consent when using embryos
Ovarian tissue cryopreservation with IVM	Preparation of the sample is done via surgery, then cryopreserved, most commonly during gender-affirming surgery	Only available option for prepubertal transgender patients No need for cessation of gender-affirming hormonal therapy and pelvic exams leading to gender dysphoria	Invasive method involving surgery Need for autologous transplantation into the pelvic cavity Not widely applied due to lack of clinical data

COS: controlled ovarian stimulation; IVM: in vitro maturation.

**Table 2 biomedicines-10-02279-t002:** Fertility preservation options for transwomen.

Fertility Preservation Method	Description	Advantages	Limitations
Sperm cryopreservation	Semen sample obtained by masturbation and cryopreserved	Well-established method Less invasive Cost-efficient Effective method for fertility preservation Partner not needed	Post-pubertal patients Cessation of hormonal treatment recommended Self-ejaculation process needed, which may lead to gender dysphoria Multiple samples may be necessary due to relatively poor quality of sperm
Testicular tissue cryopreservation	Surgically obtained testicular tissue is cryopreserved	Only available option for prepubertal patients No need to go through masturbation/ejaculation for patients with poor sperm parameters	Experimental Invasive surgery under anesthesia Costly auto-transplantation to the testis, scrotum, or ectopic subdermal locations may be required

**Table 3 biomedicines-10-02279-t003:** Ongoing studies regarding fertility issues of transgender patients.

Study Title	Identifier	Conditions	Primary Endpoints
Micro RNA Profile in the Ovarian Follicle Fluid of Transgender Men Treated with Testosterone and the Association with Fertility Potential	NCT03725280	IVF Fertility preservation Infertility	Characterization of micro RNA profile in follicular fluid of transmen who have been treated with testosterone
Fertility preservation in Transgender Persons: A Retrospective Look at the Decision and a Survey of the Desire for Children	NCT05120245	Fertility preservation in transgender persons	Transgender people’s retrospective view of the decision Satisfaction/regret about the decision Intention to use the frozen egg or sperm to fulfill a desire to have children
Fertility preservation and Reproductive Needs of Transgender People: Desires, Attitude, and Knowledge of Subjects with Gender Dysphoria (PaFer)	NCT03836027	Transgender adolescents and young adults	Reproductive desire and attitudes of transgender people toward fertility preservation
Fertility Decision Making in Youth and Young Adults (AFFRMED)	NCT05175170	Fertility preservation Parenting Gender dysphoria	Change in fertility knowledge

IVF: in vitro fertilization.

## Data Availability

Not applicable.

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
