# Peer review of "Fertility Preservation and Reproductive Potential in Transgender and Gender Fluid Population"

_biomedicines, 2022, doi:10.3390/biomedicines10092279_

Round 1

Reviewer 1 Report

This a well written review about fertility preservation in the transgender en gender fluid population.

I strongly suggest to add information on the later use of the gametes with the focus on the gender affirming treatment received, partner choice, etc. 

Author Response

This a well written review about fertility preservation in the transgender in gender fluid population.

REPLY: We are very much thankful to the reviewer for the thorough review. We agree to all specific comments addressed and have revised our paper in light of the useful suggestions. Answers to the specific comments/suggestions/queries are as follows.

I strongly suggest to add information on the later use of the gametes with the focus on the gender affirming treatment received, partner choice, etc.

REPLY: We agree that our manuscript did not detail the later use of sperm cells and oocytes in regard to gender affirming treatment and choice of partner. Transwomen who have completed genital reconstruction have irreversible sterility and there are no applicable fertility preservation methods to date. However, experimental and alternative options such as uterine transplantation may be accessible to this specific patient population. The usage of cadaveric donors may be preferred, as both vagina and uterus would be possible. However, multifactorial reasons such as anatomic and hormonal differences in transwomen, must be investigated before this option is realistically possible.  

In transmen with a male partner, ovarian tissue may be used for IVM with fertilization of mature oocytes by partner sperm and use of a gestational carrier. In transmen with a female partner, ovarian tissue may be used for IVM with fertilization of mature oocytes by donor sperm and embryo transfer to partner's uterus. Transgender and gender fluid individuals vary in elected partners, and it is important to remember that a couple hoping to expand their family may require assistance from a gestational carrier to complete their pregnancy goals.

In light of the reviewer’s comments, we have added information about viable pregnancy options in transmen and edited the gender reaffirming therapy and fertility methods sections of our manuscript. Thank you for your valuable insight.

Reviewer 2 Report

Dear authors,

the review article "Fertility-Preservation and Reproductive Potential in

Transgender and Gender Fluid Population" created by Choi and Kim deals with a very up to date topic. I enjoyed reading your  article. The style is very good and the structure is clear. I did not detect any mistake of the manuscript``s content. I know that it is a hard issue, but may you should also reflect about the issue of surrogacy for transmen after gender reassignment surgery.

I opt for publication of the manuscript.

Author Response

Dear authors,

the review article "Fertility-Preservation and Reproductive Potential in Transgender and Gender Fluid Population" created by Choi and Kim deals with a very up to date topic. I enjoyed reading your article. The style is very good and the structure is clear. I did not detect any mistake of the manuscript’s content.

REPLY: We are very much thankful to the reviewer for the thorough review. We agree to all specific comments addressed and have revised our paper in light of the useful suggestions. Answers to the specific comments/suggestions/queries are as follows.

I know that it is a hard issue, but may you should also reflect about the issue of surrogacy for transmen after gender reassignment surgery.

I opt for publication of the manuscript.

REPLY: Viable options for pregnancy and surrogacy are factors that every transgender person that undergoes gender reassignment has to consider. During discussions of oocyte and sperm retrieval, it is important discuss plans for the carrier of a future embryo. Some transmen may desire to carry a pregnancy for personal reasons or medical reasons such as infertility of the partner. Others do not desire for reimplantation into their own womb because they feel that this would not match their gender identity. Moreover, transmen who have completed both gender-affirming hormone therapy and surgery have irreversible sterility and therefore cannot achieve pregnancy. Transgender and gender fluid individuals vary in elected partners, therefore, it is essential to realize that a couple hoping to expand their family may require assistance from a gestational carrier to complete their pregnancy and family building.

On the other hand, many transgender individuals report an interest in family building via adoption and we believe that this too should be discussed in our manuscript as another outcome for surrogacy and family building and we have edited our manuscript to include the aforementioned topics of interest. Thank you for your valuable insight and comments.